# Usefulness of the Thrombotic Microangiopathy Score as a Promising Prognostic Marker of Septic Shock for Patients in the Emergency Department

**DOI:** 10.3390/jcm8060808

**Published:** 2019-06-06

**Authors:** Dong Ryul Ko, Taeyoung Kong, Hye Sun Lee, Sinae Kim, Jong Wook Lee, Hyun Soo Chung, Sung Phil Chung, Je Sung You, Jong Woo Park

**Affiliations:** 1Department of Emergency Medicine, Yonsei University College of Medicine, 06273 Seoul, Korea; kkdry@yuhs.ac (D.R.K.); grampian@yuhs.ac (T.K.); HSC104@yuhs.ac (H.S.C.); emstar@yuhs.ac (S.P.C.); 2Department of Emergency Medicine, Graduate School of Medicine, Kangwon National University, 24289 Chuncheon, Korea; 3Department of Research Affairs, Biostatistics Collaboration Unit, Yonsei University College of Medicine, 06273 Seoul, Korea; HSLEE1@yuhs.ac (H.S.L.); ksn1214@yuhs.ac (S.K.); 4Department of Laboratory Medicine, Konyang University Hospital, 35365 Daejon, Korea; lee423619@kyuh.ac.kr; 5Department of Emergency Medicine, Good Sunlin Hospital, 37725 Pohang, Korea; 6Department of Emergency Medicine, Graduate School of medicine, Kosin University, 49267 Busan, Korea

**Keywords:** sepsis, thrombotic microangiopathy score, mortality, predictor

## Abstract

The thrombotic microangiopathy (TMA) score based on the development and morphological characteristics of schistocytes is a rapid, simple biomarker that is easily obtained from the complete blood cell count by an automated blood cell analyzer. We aimed to determine whether the TMA score is associated with 30-day mortality of patients with early-stage septic shock. This observational cohort study was retrospectively conducted based on a prospective emergency department (ED) registry (June 2015–December 2016). We analyzed the TMA score at ED admission and 24 h later. The primary endpoint was all-cause mortality within 30 days of ED admission. A total of 221 patients were included. Increased TMA scores at time 0 (odds ratio (OR), 1.972; 95% confidence interval (CI), 1.253–3.106; *p* = 0.003) and at time 24 (OR, 1.863; 95% CI, 1.863–3.066; *p* = 0.014) were strong predictors of 30-day mortality. Increased predictability of 30-day mortality was closely associated with TMA scores ≥2 at time 0 (OR, 4.035; 95% CI, 1.651–9.863; *p* = 0.002) and ≥3 at time 24 (OR, 5.639; 95% CI, 2.190–14.519; *p* < 0.001). Increased TMA scores significantly predicted 30-day mortality for patients with severe sepsis and septic shock and can be helpful when determining the initial treatment strategies without additional costs or effort.

## 1. Introduction

Sepsis is defined as life-threatening acute dysfunction of the organs due to a dysregulated response to infection [1,2,3]. Although the in-hospital mortality rate has declined, it is still high (17.1%–18.0%) [2,4]. Since there are no completely effective pharmacological treatments for sepsis, early recognition, proper management, and immediate treatment with appropriate antibiotics are important and should be implemented to reduce the burden of subsequent disease [4]. To improve proper recognition and early management of sepsis, physicians should consider emergency departments (EDs) as key targets because the majority of sepsis cases are present there [4]. 

Although the Acute Physiological and Chronic Health Evaluation (APACHE) II score and the Sequential Organ Failure Assessment (SOFA) score have been commonly used for critically ill patients because of their ability to reflect disease severity and represent organ dysfunction, respectively, their rapid measurement is complicated [5]. 

The application of new biomarkers that quickly and easily predict the development of severe complications and mortality in sepsis may improve the prognosis of patients by allowing early aggressive treatment and innovative preventive and therapeutic options [6,7,8]. Since the thrombotic microangiopathy (TMA) score is a more rapid and simple biomarker that is easily, simultaneously, and automatically obtained from the complete blood cell (CBC) count by an automated blood cell analyzer, we used this score to screen patients with severe sepsis or septic shock [5]. Systemic endothelial injury develops in critically ill patients due to transplantation, autoimmune disease, cancer, cardiopulmonary bypass, and exposure to infection, radiation, and medications [5,9]. TMA is characterized by fragmented erythrocytes (schistocytes), thrombocytopenia, and higher levels of lactate dehydrogenase [10]. When diagnosing TMA, an important criterion is the presence of ≥1% schistocytes on a peripheral blood smear [10]. Moreover, the development of schistocytes in the blood reflects the possibility of thrombocytopenia-associated multiple organ failure (TAMOF) [9]. 

The TMA score was developed to screen for TAMOF in critically ill patients; TAMOF is determined based on the development and morphological characteristics of schistocytes [9]. However, schistocytes are occasionally produced by severe hemolysis in turbulent areas of the microcirculation partially occluded by platelet aggregations [11,12]. Schistocytes can be characterized by microcytic hyperchromic red blood cells (RBCs), increased red cell distribution width (RDW), and increased hemoglobin distribution width (HDW). The TMA score can be used for early detection of TAMOF based on RBC parameters and the volume/hemoglobin concentration (V/HC) determined by an automated hematology analyzer [9].

Similar to classic TMA, such as thrombotic thrombocytopenic purpura (TTP) and hemolytic-uremic syndrome (HUS), a systemic response to severe infection and sepsis can induce disseminated intravascular coagulation (DIC). With DIC, the activated cascade of coagulation causes thrombi formation due to excessive fibrin in the intravascular microcirculation [13]. The excessive fibrin formation induces thrombocytopenia and clotting factor consumption. Mechanical damage of the RBCs by excess fibrin strands also leads to the formation of schistocytes [13]. Considering this information, we designed this study to examine the development and morphological changes of schistocytes during severe and septic shock. To the best of our knowledge, no studies have focused on the advantages of using only the TMA score to predict short-term mortality of patients with sepsis. We hypothesized that the TMA score could be used to predict the severity of sepsis and eventual mortality for patients with severe and septic shock. 

Additionally, lactate levels have been used clinically as a reliable indicator of the severity and outcome of sepsis or septic shock because they traditionally reflect tissue hypoxia and can be measured within a short time after ED admission using point-of-care testing [14,15]. Therefore, we aimed to determine whether the initial TMA score was significantly associated with 30-day mortality for patients with early-stage septic shock and whether combining the initial lactate concentrations with the TMA score improved the ability to predict unfavorable outcomes.

## 2. Experimental Section

### 2.1. Study Population

This observational cohort study was retrospectively conducted based on a prospective registry of the ED of Severance Hospital, which is affiliated with Yonsei University College of Medicine and is a tertiary-level referral hospital with a population of approximately 85,000 patients per year between June 2015 and December 2016. This study was reviewed and approved by the Institutional Review Board (3-2017-0060) of the Yonsei University Health System. The requirement for written informed consent from patients was waived 1) because of the use of the prospective early goal-directed therapy (EGDT)/SEPSIS registry of the ED and 2) because this study retrospectively analyzed data based on de-identified medical records of the subjects.

Since November 2007, the EGDT, which has been referred to as SEPSIS since June 2015 because of the changes in the EGDT, has been implemented as a critical pathway (CP) program at our institution as part of a quality improvement initiative [6]. To provide appropriate treatment for sepsis, we designed a standard treatment with bundled management to reduce unnecessary delays in the hospital by implementing rapid decision-making by specialists [6,16]. When a patient with suspicious signs of infection arrives at the ED, emergency physicians identify whether that patient is a candidate for this program as soon as possible by using predetermined guidelines based on systemic inflammatory response syndrome (SIRS). We included patients who presented with two SIRS criteria and one CP criterion (hypotension or high lactate value) as a result of infection at the time of ED admission [6,16]. We classified patients with suspected infections admitted to the ED as presenting with SIRS [6,16]. As a result of infection, we defined SIRS by the presence of two or more of the following criteria [6,16]: Temperature <36 °C or >38 °C; heart rate >90 beats/min; respiratory rate >20 breaths/min or PaCO_2_ <32 mmHg; and white blood cell (WBC) count >12,000 cells/μL or <4000 cells/μL or band neutrophil count >10% [6,16]. Then, we assessed the patient’s eligibility for the CP program. Those who presented with SIRS and had at least one of the following inclusion criterion at the time of ED admission were finally included in the CP program: Systolic blood pressure <90 mmHg despite a 20- to 30-mL/kg intravenous crystalloid fluid challenge or serum lactate level ≥4 mmol/L at the time of ED admission [6,16]. The following cases were excluded or deactivated from the CP program: Age younger than 18 years; pregnancy; acute cerebrovascular or coronary syndrome; active gastrointestinal bleeding; contraindication to central venous catheter; trauma; necessity for immediate surgery; transfer to another hospital within six hours after ED admission; and do not resuscitate status. We included consecutive patients who were prospectively integrated in the CP program for sepsis treatment based on predetermined protocols [6]. Additionally, we excluded patients with prosthetic valves, patients who received a transplant, and patients undergoing chemotherapy. 

### 2.2. Data Collection 

The following demographic data were collected from the registry: Age; sex; body weight; medical history, including hypertension, diabetes mellitus, cardiovascular disease, heart failure, chronic kidney disease, cancer, and chronic liver disease; hemodynamic parameters; laboratory results, including CBC count, routine chemistry, C-reactive protein (CRP), lactate, albumin, procalcitonin, and blood culture results; in-hospital course, including time of antibiotics initiation; clinical outcomes; and follow-up period. To evaluate the clinical severity of each patient’s case, the SOFA score was determined using the worst values obtained during the initial 24 h of the ED admission. 

### 2.3. Assessment of TMA Score 

The TMA score ranges from 0 to 5 points, with 1 point given for each of the following: RDW >15%, HDW >3.2%, percentage of microcytes ≥0.4% (percentage of microcytes indicates the percentage of RBCs smaller than 60 fL), percentage of hyperchromic red cells ≥1.9% (percentage of hyperchromic red cells indicates the percentage of RBCs with more than 41 g/dL of hemoglobin), and platelet count <140 × 10^9^/L [9]. All these RBC parameters were measured using an automated blood cell analyzer (ADVIA 120, Siemens, Forchheim, Germany) [9]. The percentages of microcytes and hyperchromic red cells were calculated using the V/HC cytogram [9]. The TMA score was automatically calculated and reported [9]. To investigate the usefulness of TMA to predict the development of short-term mortality, we analyzed the TMA scores at the time of ED admission and 24 h later. We collected venous blood of the subject in ethylenediaminetetraacetic-containing vacutainers within 15 min after ED admission (time 0) and within 24 ± 6 h after admission (time 24). 

### 2.4. Clinical Endpoints 

The primary endpoint was the all-cause mortality within 30 days of ED admission. 

### 2.5. Statistical Analysis 

Demographic and clinical data are represented as medians and interquartile ranges (IQRs), means and standard deviations, percentages, or frequencies, as appropriate. Continuous variables were compared using the two-sample *t*-test or Mann-Whitney U-test, and categorical variables were compared using the *χ*^2^ test or Fisher’s exact test. To identify the significance of the differences between the groups over time, a linear two-factor mixed model was conducted using a repeated-measures covariance pattern and unstructured covariance within patients. We used two fixed effects for this model: The time effect (TMA at time 0 and time 24) and the diagnostic effect for the clinical endpoint (development and non-development of 30-day mortality). Univariable analyses were performed to identify the significance of the relationships among the demographic and clinical data. To highlight the predictor of 30-day mortality, variables with *p* < 0.05 were selected from our univariable analysis and the promising predictors of 30-day mortality during the early stage of sepsis were clarified using multivariable logistic regression analysis with a stepwise approach. Consequently, the results are shown for forward conditional stepwise selections of clinical markers with *p* < 0.05 for entry and *p* < 0.05 for retention. We created receiver-operating characteristic (ROC) curves and determined the area under the curve (AUC) to identify the ability of the TMA score to predict 30-day mortality. We performed Youden’s method to verify the optimal cutoff of the delta neutrophil index for discriminating between the development and non-development of 30-day mortality. These results are represented as odds ratios (ORs) and 95% confidential intervals (CIs). After considering the development of 30-day mortality and its AUC, we compared the diagnostic performances of the TMA score and each marker, including WBC, CRP, lactate, and procalcitonin, on ED admission. Based on the optimal cutoff values of 2 for TMA on ED admission and of 3 for TMA at time 24, Kaplan–Meier survival curves were created using 7-day and 30-day mortality data, and the groups were compared using the log-rank test.

In addition, to increase the early predictability of 30-day mortality during the ED admission, we analyzed the ROC curve of lactate on ED admission by combining the TMA score over time and the lactate value on admission. Based on the optimal cutoff values of 2 for TMA on ED admission, of 3 for TMA at time 24, and of 4 for lactate with regard to the sepsis guidelines, we classified three ED admission groups and three time 24 groups. The ED admission groups were based on optimal cutoff values of 2 for TMA and 4 for lactate: Group 1, lactate <4 and TMA <2; group 2, lactate ≥4 and TMA <2 or lactate <4 and TMA ≥2; and group 3, lactate ≥4 and TMA ≥2. The time 24 groups were based on an optimal cutoff value of 3 for TMA and 4 for lactate: Group 1, lactate <4 and TMA <3; group 2, lactate ≥4 and TMA <3 or lactate <4 and TMA ≥3; and group 3, lactate ≥4 and TMA ≥3. We identified sensitivity and specificity according to groups. Sensitivity, specificity, positive predictive values, and negative predictive values of the TMA score, lactate concentrations, and a combination of the TMA score and lactate concentrations for predicting 30-day mortality were defined. Statistical analyses were performed using SAS version 9.2 (SAS Institute Inc., Cary, NC) and MedCalc Statistical Software version 16.4.3 (MedCalc Software bvba, Ostend, Belgium), and *p* < 0.05 was considered significant.

## 3. Results

### 3.1. Study Population, Clinical Evaluation, and Treatment

Figure 1 shows the inclusion and clinical outcomes of subjects with septic shock registered in the EGDT/SEPSIS program (Figure 1). A total of 204 patients were included in this study, and 31 patients experienced 30-day mortality after ED admission (Table 1). The incidence of the all-cause 30-day mortality was 15.2%. Ninety-one of them were male (44.61%), and the mean age was 67 ± 15 years. The mortality group (29.03%) showed higher rates of cardiovascular disease than did the survival group (13.87%, *p* = 0.035). The SOFA score was significantly increased in patients who died within 30 days (10.39 ± 2.93), as compared with patients who survived (7.11 ± 2.53, *p* < 0.001). There were significant differences for lactate, albumin, and total CO_2_ between the two groups (*p* of all <0.001). TMA score at time 0 and 24 were significantly increased in patients who died as compared to patients who survived (1.42 ± 1.02 and 1.69 ± 1.16 in the survival group versus 2.23 ± 1.15 and 2.82 ± 1.30 in the mortality group, *p* of all <0.001). Patients who experienced 30-day mortality had significantly higher TMA scores than those who did not. Mean admission to antibiotics time was 3.64 h (SD 4.54), and this was significantly shorter in patients who died within 30 days compared to those who did not (2.70 ± 1.44 vs. 3.81 ± 4.88, *p* = 0.016). The linear mixed model revealed significant differences in the TMA scores between patients grouped according to the development and non-development of 30-day mortality (group, *p* < 0.001; time, *p* < 0.001; and group × time, *p* = 0.007) after ED admission (Figure 2A).

### 3.2. TMA Score as a Predictor of 30-Day Mortality for Sepsis

The ROC curve analysis of the prediction of 30-day mortality indicated that the AUCs regarding the TMA score at time 0 and time 12 were 0.697 (IQR, 0.589–0.789; *p* < 0.001) and 0.738 (IQR, 0.6–0.853; *p* < 0.001), respectively (Figure 2B,C). The univariable analysis showed significant differences in the TMA scores at different time points for patients who did and did not develop 30-day mortality (Appendix A). Further multivariable logistic regression analyses demonstrated that increased TMA scores at time 0 (OR, 1.972; 95% CI, 1.253–3.106; *p* = 0.003) and time 24 (OR, 1.863; 95% CI, 1.863–3.066; *p* = 0.014) were strong predictors of 30-day mortality (Table 2). The optimal cut-off values of the TMA scores at time 0 and time 24 were 2 (sensitivity, 77.4 (IQR, 62.7–92.1); specificity: 54.1 (IQR, 46.6–61.5)) and 3 (sensitivity: 63.6 (IQR, 43.5–83.7); specificity: 76.3 (IQR, 69.6–83.1)), respectively (Figure 2B,C). The increased predictability of 30-day mortality in the present study was closely associated with a TMA score ≥2 at time 0 (OR, 4.035; 95% CI: 1.651–9.863; *p* = 0.002) and a TMA score ≥3 at time 24 (OR, 5.639; 95% CI: 2.190–14.519; *p* < 0.001; Figure 2B,C). Higher TMA scores at admission and 24 h after admission were significantly associated with an increased risk of 30-day and 7-day mortality among patients with septic shock (Figure 3A,B and Appendix A).

### 3.3. Comparison of the TMA Score and Conventional Clinical Markers as Predictors of Mortality for Patients with Sepsis

Comparisons of the ROC curves performed to predict the development of 30-day mortality showed that the AUC for the TMA score at ED admission was significantly superior to CRP. The AUC for the TMA score at time 24 was significantly superior to those for CRP and procalcitonin. Moreover, the TMA scores at ED admission and time 24 were not significantly inferior to SOFA scores within 24 h and lactate at ED admission (Figure 4 and Appendix A).

### 3.4. Prognostic Value of the TMA Score in Combination with Lactate on ED Admission

To improve the predictability of clinical outcomes early after ED admission, we demonstrated how predictability was improved when the TMA score was added to the lactate level on ED admission. The AUC of the addition of the TMA score at time 0 or time 24 to the lactate level (0.857 (*p* = 0.034) and 0.892 (*p* = 0.018)) on ED admission was significantly improved compared to the AUC of the lactate level alone (0.8; IQR, 0.703–0.878; Figure 5 and Appendix A). We obtained sensitivity and specificity using a combination of the optimal cutoff of the TMA score and lactate level. For those in the ED admission groups based on optimal cut-off values of 2 for TMA and 4 for lactate, the sensitivity and specificity for patients with lactate ≥4 and TMA <2 or lactate <4 and TMA ≥2 were 96.8 (IQR, 90.6–100.0) and 41.2 (IQR, 33.8–48.6). Sensitivity and specificity for those with lactate ≥4 and TMA ≥2 were 45.2 (IQR, 27.6–62.7) and 88.8 (IQR, 84.1–93.6). For those in the time 24 groups based on optimal cut-off values of 3 for TMA and 4 for lactate, the sensitivity and specificity for lactate ≥4 and TMA <3 or lactate <4 and TMA ≥3 were 90.9 (IQR, 78.6–100.0) and 61.3 (IQR, 53.5–69.1). Sensitivity and specificity for those with lactate ≥4 and TMA ≥3 were 40.9 (IQR, 20.4–61.5) and 92.7 (IQR, 88.5–96.8; Table 3).

## 4. Discussion

In the present study, sepsis patients who died within 30 days after ED admission had significantly higher TMA scores than did those who survived. The higher TMA scores at time 0 and time 24 were significant predictors of 30-day mortality. Sepsis is a clinically complex condition that is characterized by a systemic inflammatory response accompanying an infection [6,17]. Although early recognition and treatment of sepsis has significantly decreased its mortality rate over the years, it remains high.

The pathophysiology of sepsis is very complex and poorly understood. Among various mechanisms of sepsis-induced organ failure, sepsis is significantly associated with the activation of the coagulation cascade that sometimes leads to DIC, which is its more severe clinical form [18]. Although the classic forms of TMA are HUS and TTP, several different forms such as DIC, atypical hemolytic uremic syndrome, scleroderma renal crisis, malignant hypertension, antiphospholipid antibody syndrome, and drug toxicities are considered other conditions of TMA [19]. The schistocyte count is not a clue to the initial diagnosis of DIC in sepsis and is not completely specific to TMA. Nevertheless, the International Council for Standardization in Hematology (ICSH) recommended that the presence of >1% schistocytes on a peripheral blood smear without other moderate RBC changes is an important criterion for the diagnosis of TMA; furthermore, Lesesve et al. suggested that the presence of >1% schistocytes was frequently related to various circumstances in which the increased schistocyte count indicated severe infection, pregnancy, and leukemia [10,20,21]. Most patients with DIC after septicemia had schistocytes, although only approximately 0.5% were seen. Visudhiphan et al. proposed that there was no difference in schistocyte counts of sepsis patients with and without DIC [22]. Lesesve et al. suggested that it is difficult to differentiate schistocytes that originated from DIC because sepsis itself and renal impairment caused by sepsis can create schistocytes in patients with sepsis [21].

Considering the pathogenesis of severe sepsis and septic shock and the significant association between sepsis and DIC, increased schistocyte development may be significantly associated with the severity of sepsis and septic shock. Moreover, Bateman et al. proposed that the mechanisms of erythrocytes during sepsis are changed by the increased rigidity and decreased deformability of RBCs [23]. These changes in erythrocytes are clinically relevant for sepsis patients [23]. First, decreased erythrocyte deformability develops early, within 24 h of admission of critically ill patients, and does not recover in septic patients. This RBC deformability progressively decreases over the next two to eight days in patients with infections and is significantly associated with organ dysfunction and the outcomes of sepsis patients [23,24]. Therefore, RBC injury in sepsis can lead to microvascular dysregulation and mechanical damage and has a significant role in the formation of schistocytes [23].

In the present study, the TMA score ranged from 0 to 5 points and was used based on simple changes similar to the morphological characteristics of schistocytes to evaluate whether increased TMA scores can predict the severity of sepsis and the development of 30-day mortality for patients with septic shock after ED admission [9]. The optimal cutoff values of TMA scores at time 0 and time 24 were 2 and 3. We demonstrated that TMA scores ≥2 at time 0 and ≥3 at time 24 increased the predictability of 30-day mortality for patients with sepsis. The score can be very meaningful during the early stage of sepsis because we assessed the changes in schistocytes rather than the absolute schistocyte counts. The TMA score is automatically calculated and reported by an automated hematology analyzer when performing a CBC count, and no additional costs or laboratory tests are necessary [5,9]. It is a simple and rapidly measurable marker [5]; therefore, our results demonstrated that the TMA scores for those with sepsis could be a useful ancillary marker. Several studies revealed the usefulness of the automated counting of RBC fragments as a routine screening tool, good agreement with microscopy, and other advantages [10,20].

There was no significant increase in the predictability of 30-day mortality with the addition of TMA scores to SOFA scores, but the addition of TMA scores to lactate levels on ED admission could significantly increase the predictability of 30-day mortality. Similar to the TMA score, which can be obtained simultaneously with CBC analysis results, lactate levels can be obtained soon after ED admission using point-of-care testing. A high concentration of serum lactate was significantly associated with higher in-hospital mortality rates and 90-day mortality rates compared to isolated refractory hypotension among septic patients [25,26]. Gotmaker et al. demonstrated that patients with an isolated blood lactate level >4 mmol/L were at greater risk for mortality than those with isolated refractory hypotension [26]. They also proposed that a lactate level >4 mmol/L should be considered a key risk stratification tool even in the absence of refractory hypotension in patients with suspected or proven infection after ED admission [26]. Based on the optimal cutoff TMA scores for ED admission and time 24 and the critical value of lactate on ED admission, we designed a simple, intuitive, and categorical predictor with good sensitivity and specificity. When the optimal cutoff values of TMA scores at time 0 and time 24 were 2 and 3, sensitivity values were 77.4 (IQR, 62.7–92.1) and 63.6 (IQR, 43.5–83.7) and specificity values were 54.1 (IQR, 46.6–61.5) and 76.3 (IQR, 69.6–83.1). If only one of the cutoff values for the TMA score or lactate level is exceeded in patients with sepsis, then the sensitivity of the prediction of 30-day mortality is significantly increased. If both cutoff values for the TMA score and lactate level are exceeded, then the specificity is significantly increased. The TMA score in the ED may be useful to predict 30-day mortality in patients with severe sepsis and septic shock. Combining TMA score with lactate concentrations improves the predictive performance for the prognosis and may contribute to rapid risk stratification of patients with severe sepsis and/or septic shock.

The TMA score reflects the development of schistocytes and implies mechanical hemolysis by partial occlusion of the microcirculation, and increased lactate values suggest deterioration of tissue hypoxia caused by microcirculation dysfunction. Therefore, the TMA score is helpful when making a clinical decision during the early stage of sepsis.

The present study had several limitations. Although this study included a prospective CP using a predetermined standardized protocol, it was performed retrospectively using a cohort derived from the CP of a single, tertiary, academic hospital. Therefore, the design of this study could have caused a selection bias because it was difficult to control confounding factors. Since January 2019, the EGDT/SEPSIS as a CP protocol was revised to new SEPSIS according to the third international consensus definitions for sepsis and septic shock (Sepsis-3) [27]. However, because of the timing of the study period, we did not include septic patients enrolled according to the new sepsis CP. We relied on the explicit Sepsis-2 to identify and enroll sepsis and septic shock cases in the registry [28]. Further prospective, multicenter trials are required to validate the usefulness of the TMA scores as a prognostic marker for severe sepsis or septic shock as defined by their third international consensus definitions.

The TMA scores in the present study ranged from 0 to 5 points and were designed based on the development of schistocytes and changes similar to morphological characteristics of schistocytes. We could not directly compare the correlation between automatically measured TMA scores and the schistocyte percent estimated by microscopically counting.

Finally, despite the use of a prospective registry as the CP, we could not measure ADAMTS-13 activity, which is known as an important factor in the pathophysiology of the classic form of TMA, because it was not a mandatory biomarker. However, the objective of this study was to investigate the severity of sepsis by detecting the development of schistocytes using automatic measurements.

## 5. Conclusions

We found that increased TMA scores predicted the 30-day mortality of patients with severe sepsis and septic shock. Combining TMA score with initial lactate concentrations can improve the predictive performance of short-term mortality. The TMA score is automatically measurable by an automated hematology analyzer. It can be quickly measured without additional costs or effort; therefore, the TMA score may be a promising tool for rapid risk stratification of patients with severe sepsis and/or septic shock in need of intensive care and innovative therapies.

## Figures and Tables

**Figure 1 jcm-08-00808-f001:**
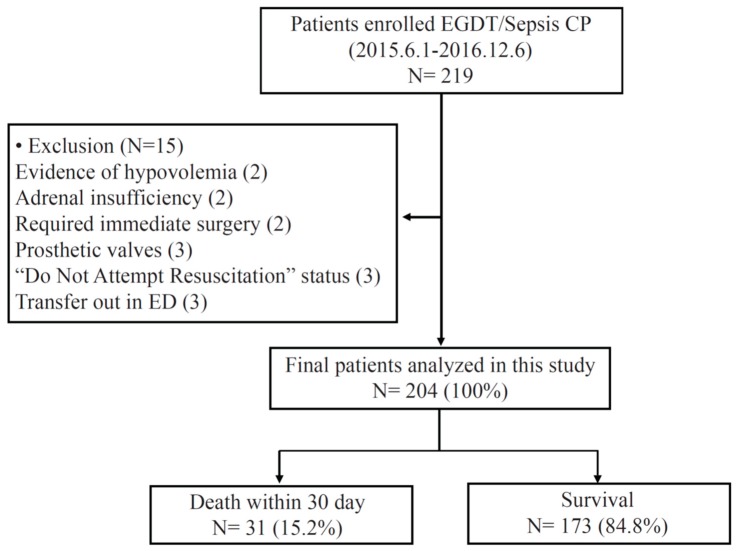
Flow diagram of patient inclusion.

**Figure 2 jcm-08-00808-f002:**
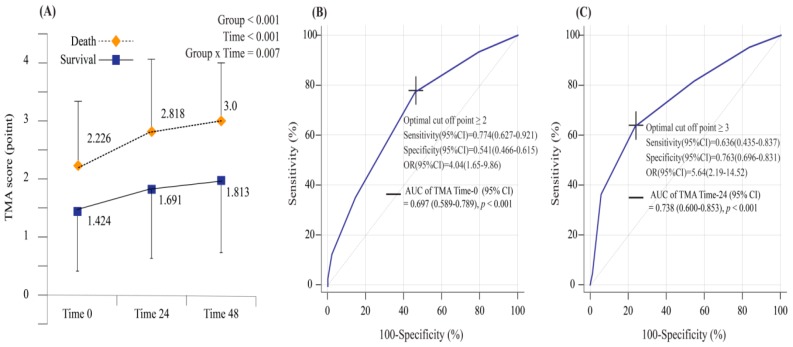
(**A**) Linear mixed model of the thrombotic microangiopathy (TMA) score to estimate significant differences between groups over time according to 30-day mortality. The receiver-operating characteristic curves for predictability of the TMA score at admission (**B**) and 24 h (**C**) after admission according to 30-day mortality are shown.

**Figure 3 jcm-08-00808-f003:**
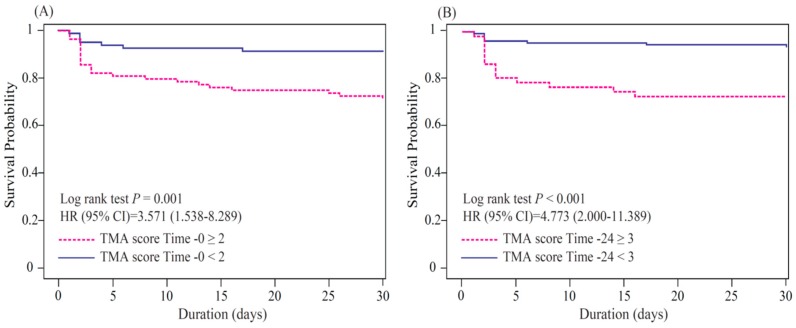
The thrombotic microangiopathy (TMA) score as a predictor of 30-day mortality. Higher TMA scores at admission (**A**) and 24 h (**B**) after admission were significantly associated with an increased risk of 30-day mortality among patients with septic shock.

**Figure 4 jcm-08-00808-f004:**
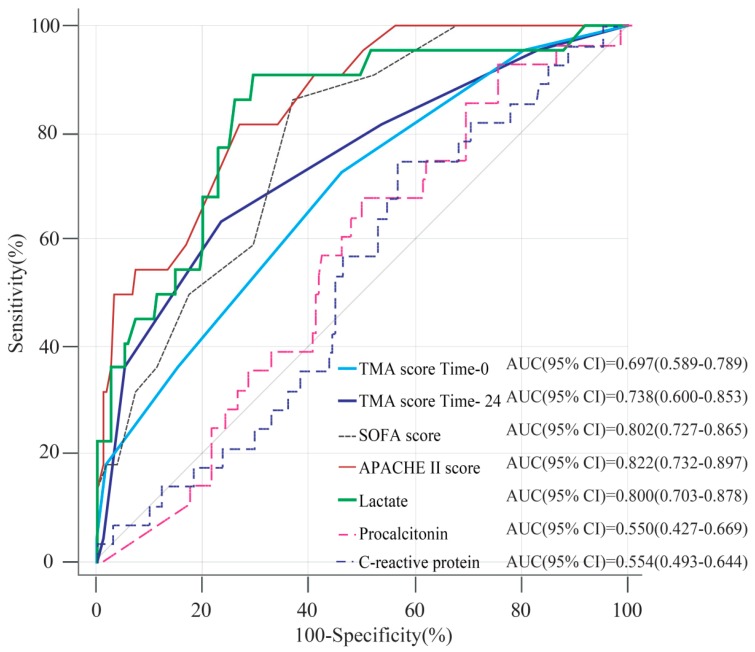
Comparison of the area under the curve (AUC) for the thrombotic microangiopathy (TMA) score when predicting the 30-day mortality (A). The AUC showed discriminative abilities for risk stratification of 30-day mortality. Statistical information is shown in Appendix A. The AUC of TMA at 24 h was statistically superior to the C-reactive protein and procalcitonin concentration, and it was not inferior to the Sequential Organ Failure Assessment (SOFA) score, and lactate concentration according to the early goal-directed therapy (EGDT)/SEPSIS critical pathway

**Figure 5 jcm-08-00808-f005:**
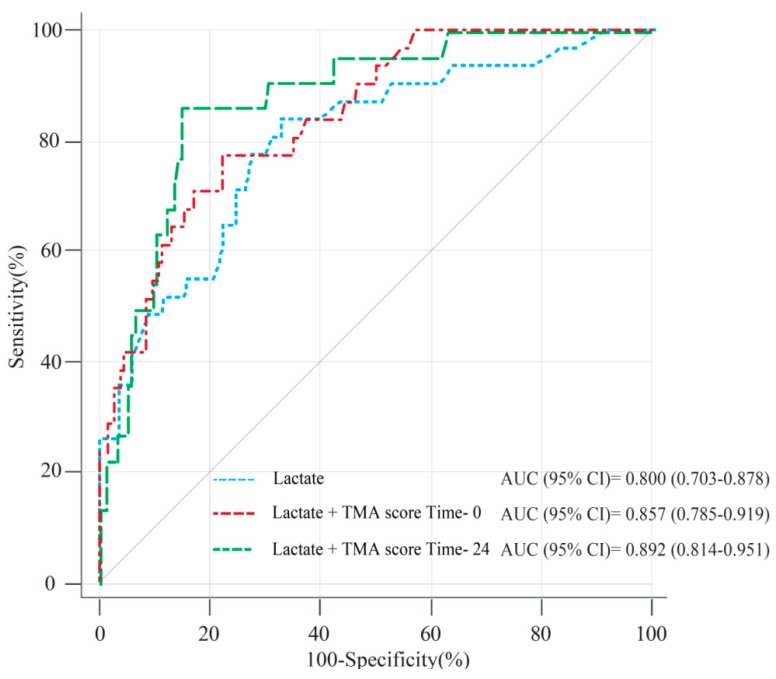
Receiver-operating characteristic curve of lactate and the thrombotic microangiopathy (TMA) score for the 30-day mortality of patients with septic shock.

**Table 1 jcm-08-00808-t001:** Clinical characteristics of patients stratified by 30-day mortality.

Variables	Total	Survival	Death	*p*
	*N* = 204 (100%)	*N* = 173 (84.8%)	*N* = 31 (15.2%)	
Age (years)	67 ± 15	66 ± 15	72 ± 11	0.016 *
Male sex (*n* (%))	91 (44.61)	78 (45.09)	13 (41.94)	0.745
BMI (kg/m^2^)	21.47 ± 3.43	21.41 ± 3.36	21.82 ± 3.83	0.542
SOFA score (points)	7.61 ± 2.84	7.11 ± 2.53	10.39 ± 2.93	<0.001 *
Initial Vital Sign				
Systolic blood pressure (mmHg)	85.46 ± 24.03	86.73 ± 23.39	78.32 ± 26.61	0.073
Diastolic blood pressure (mmHg)	52.93 ± 15.10	53.83 ± 15.2	47.90 ± 13.72	0.044 *
Heart rate (bpm)	104.46 ± 24.72	104.10 ± 25.70	106.45 ± 18.55	0.545
Respiratory rate (bpm)	18.59 ± 3.72	18.57 ± 3.82	18.71 ± 3.19	0.844
Body temperature (°C)	37.51 ± 3.03	37.83 ± 1.32	35.719 ± 6.94	0.102
Comorbidity (*n* (%))				
Hypertension	121 (59.31)	102 (58.96)	19 (61.29)	0.808
Diabetes mellitus	66 (32.35)	52 (30.06)	14 (45.16)	0.098
Cardiovascular disease	33 (16.18)	24 (13.87)	9 (29.03)	0.035 *
Heart failure	7 (3.43)	6 (3.47)	1 (3.23)	>0.999
Chronic kidney disease	23 (11.27)	21 (12.14)	2 (6.45)	0.54
Liver disease	23 (11.27)	19 (10.98)	4 (12.90)	0.759
Malignancy	52 (25.49)	43 (24.86)	9 (29.03)	0.623
Treatment				
Admission to antibiotics time (hours)	3.64 ± 4.54	3.81 ± 4.88	2.70 ± 1.44	0.016 *
Admission to vasopressor time (hours)	2.63 ± 3.92	2.76 ± 4.20	1.92 ± 1.59	0.052
Antibiotics administration <3 h (*n* (%))	135 (66.18)	111 (64.16)	24 (77.42)	0.151
Laboratory Data				
White blood cell count (10^3^/μL)	13.88 ± 9.25	14.16 ± 8.65	12.33 ± 12.15	0.429
Hematocrit (%)	35.88 ± 6.52	36.10 ± 6.04	34.62 ± 8.73	0.37
Platelet count (10^3^/μL)	185 ± 106	188 ± 98	169 ± 144	0.486
Neutrophil count (10^3^/μL)	12.40 ± 8.77	12.70 ± 8.20	10.72 ± 11.49	0.363
Prothrombin time (INR)	1.25 ± 0.39	1.23 ± 0.40	1.37 ± 0.34	0.069
Creatinine (mg/dL)	2.07 ± 1.65	1.99 ± 1.62	2.54 ± 1.78	0.088
C-reactive protein (mg/L)	160 ± 117	158 ± 116	175 ± 121	0.459
Procalcitonin (ng/mL)	28.60 ± 36.26	28.25 ± 36.71	30.67 ± 34.08	0.745
Albumin (g/dL)	3.14 ± 0.66	3.21 ± 0.61	2.73 ± 0.77	<0.001 *
Lactate (mmol/L)	3.53 ± 3.22	2.87 ± 2.23	7.14 ± 5.02	<0.001 *
Total CO2 (mmol/L)	17.45 ± 4.39	18.18 ± 3.86	13.23 ± 4.95	<0.001 *
Bacteremia (*n* (%))	102 (50.00)	83 (47.98)	19 (61.29)	0.172
TMA score Time 0 (points)	1.55 ± 1.08	1.42 ± 1.02	2.23 ± 1.15	<0.001 *
TMA score Time 24 (points)	1.83 ± 1.23	1.69 ± 1.16	2.82 ± 1.30	<0.001 *

* *p* < 0.05; BMI, body mass index; SOFA, sequential organ failure assessment; TMA, thrombotic microangiopathy.

**Table 2 jcm-08-00808-t002:** Multivariable logistic regression analysis for predictors of 30-day mortality.

**Variable**	**Enter Method**			
	**OR (95% CI)**	***p***	**OR (95% CI)**	***p***
SOFA score (per 1 point)	1.237 (1.012–1.513)	0.038 *	1.184(0.949–1.476)	0.134
History of cardiovascular disease	2.730 (0.818–9.108)	0.102	2.831(0.687–11.671)	0.15
Albumin (g/dL)	0.515(0.225–1.179)	0.116	0.484(0.198–1.182)	0.111
Total CO2 (per 1 mmol/L)	0.898(0.767–1.052)	0.184	0.918(0.762–1.106)	0.368
Lactate (per 1 mmol/L)	1.247(1.039–1.497)	0.018 *	1.298(1.047–1.609)	0.018 *
TMA score Time 0 (per 1 point)	1.928(1.187–3.130)	0.008 *		
TMA score Time 24 (per 1 point)			1.683 (0.996–2.844)	0.052
**Variable**	**Stepwise Selection**			
	**OR (95% CI)**	***p***	**OR (95% CI)**	***p***
SOFA score (per 1 point)	1.264 (1.062–1.505)	0.008 *		
History of cardiovascular disease				
Albumin (g/dL)			0.410 (0.178–0.945)	0.036 *
Total CO2 (per 1 mmol/L)				
Lactate (per 1 mmol/L)	1.338 (1.156–1.548)	<0.001 *	1.430 (1.207–1.693)	<0.001 *
TMA score Time 0 (per 1 point)	1.972 (1.253–3.106)	0.003 *		
TMA score Time 24 (per 1 point)			1.863 (1.132–3.066)	0.014 *

* *p* < 0.05; OR, odds ratio; CI, confidence interval; SOFA, sequential organ failure assessment; TMA, thrombotic microangiopathy.

**Table 3 jcm-08-00808-t003:** Sensitivity, specificity, positive predictive value, and negative predictive value of the TMA score, lactate, and a combination of lactate with the TMA score for predicting the 30-day mortality of patients with septic shock.

Variable	Sensitivity (%)	Specificity (%)	PPV (%)	NPV (%)	OR (95% CI)
TMA score 0 ≥ 2	77.4 (62.7–92.1)	54.1 (46.6–61.5)	23.3 (15.1–31.5)	93.0 (88.0–98.0)	4.035 (1.651–9.863)
TMA score 24 ≥ 3	63.6 (43.5–83.7)	76.3 (69.6–83.1)	28.0 (15.6–40.4)	93.5 (89.2–97.9)	5.639 (2.190–14.519)
Lactate ≥ 4.0	64.5 (47.7–81.4)	75.4 (69.0–81.9)	32.3 (20.6–43.9)	92.1 (87.7–96.6)	5.584 (2.474–12.603)
TMA score 0 ≥ 2 and Lactate ≥ 4.0	45.2 (27.6–62.7)	88.8 (84.1–93.6)	42.4 (25.6–59.3)	89.9 (85.3–94.4)	6.545 (2.788–15.362)
TMA score 24 ≥ 3 and Lactate ≥ 4.0	40.9 (20.4–61.5)	92.7 (88.5–96.8)	45.0 (23.2–66.8)	91.4 (87.0–95.9)	8.748 (3.066–24.960)
TMA score 0 ≥ 2 or Lactate ≥ 4.0	96.8 (90.6–100.0)	41.2 (33.8–48.6)	23.1 (15.8–30.3)	98.6 (95.9–100.0)	20.998 (2.798–157.588)
TMA score 24 ≥ 3 or Lactate ≥ 4.0	90.9 (78.9–100.0)	61.3 (53.5–69.1)	25.6 (16–35.3)	97.9 (95.0–100.0)	15.862 (3.574–70.398)

AUC, area under curve; CI, confidence interval; NPV, negative predictive value; OR, odds ratio; PPV, positive predictive value; TMA, thrombotic microangiopathy.

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
