# Peer review of "Usefulness of the Thrombotic Microangiopathy Score as a Promising Prognostic Marker of Septic Shock for Patients in the Emergency Department"

_jcm, 2019, doi:10.3390/jcm8060808_

Round 1
Reviewer 1 Report
In this meta-analysis manuscript, Ko et al. aimed to determine whether the Thrombotic microangiopathy (TMA) score is independently associated with 30-day mortality of patients with early-stage septic shock. Overall, this is a straightforward, aim-derived, single center, retrospective cohort study. The authors included 221 patients in the EGDT/SEPSIS registry of the ED. With the combination of ROC curves and AUC analysis, univariate/multivariable logistic regression, the authors concluded that increased TMA scores independently predicted 30-day mortality for patients with severe sepsis and septic shock. While the conclusion has considerable significance for clinical reference, this manuscript needs to be revised based on the following concerns:
1) The authors did not declare clearly why they choose 30-day mortality as primary endpoint. From the data in figure 3, the most difference of mortality events occurred within 5 days of ED either in TMA-0 >= 2 at (A) and TMA-24 >= 3 (B), while the mortality percentage during later time course (Day 5 to 30) seems less significant than the control. Thus, this reviewer doubted whether TAM is merely a predictor of acute mortality of septic shock. Please discuss.
2) The author highlighted the “independence” of TAM in predicting 30-day mortality both in abstract and main text. However, the author also used TAM combined with other variables e.g. SOFA or lactate to compare the sensitivity and specificity, especially when add to lactate levels could significantly increase the predictability. Thus, from the future practically perspective, such highlight is not necessary.
3) Page 4, line 185. The author cited table 1 only once in the main text with only a rough description (line 184-185). Actually, there are more than TAM showed statistic significance in Table one, e.g. cardiovascular disease, albumin, lactate, total CO2. Please describe the date in the main text more detail and discuss if necessary, while please do not leave the readers to guess where the description for these data are.
4) Page 7, table 2. First, it is not clear what’s the difference between the left half panel and right half panel, because the top parameters are the same. Second, the format needs to be modified, the p value should be in the same line, it’s quite ambiguous now.
5) Please explain why the ROC curve of TAM-0 in B and TAM-24 in C where totally opposite with the one in Figure 4.
6) Page 9, line 240-242. “The AUC of the addition of the TMA score at time-0 and time-24 to the lactate level (0.8; IQR, 0.703-0.878)” this value should be replaced by the later one in line 242.
7) In the conclusion, the authors indicated that it can be helpful when determine the initial treatment strategies with the use of TAM scores. However, no such discussions were presented. What will the initial treatment strategies be different with TAM scores in the future?
Minor issues:
1) Page 2, line 82-85, lack of reference after “lactate levels have been used clinically as a reliable indicator … using point-of care testing”.
2) Since the author compared TMA score with CRP in line 224-225, page 8. Please add CRP ROC cure in Figure 4.
Author Response
** Please, check attached file for reviewing response to reviewer's comments **
Type of the Paper (Article)
Usefulness of the Thrombotic Microangiopathy Score as a Promising Prognostic Marker of Septic Shock for Patients in the Emergency Department
Response to reviewer #1’s comments
Comments and Suggestions for Authors
In this meta-analysis manuscript, Ko et al. aimed to determine whether the Thrombotic microangiopathy (TMA) score is independently associated with 30-day mortality of patients with early-stage septic shock. Overall, this is a straightforward, aim-derived, single center, retrospective cohort study. The authors included 221 patients in the EGDT/SEPSIS registry of the ED. With the combination of ROC curves and AUC analysis, univariate/multivariable logistic regression, the authors concluded that increased TMA scores independently predicted 30-day mortality for patients with severe sepsis and septic shock. While the conclusion has considerable significance for clinical reference, this manuscript needs to be revised based on the following concerns:
Response: Thank you for these comments and useful suggestions regarding our manuscript. We have responded to the comments in detail below.
1) The authors did not declare clearly why they choose 30-day mortality as primary endpoint. From the data in figure 3, the most difference of mortality events occurred within 5 days of ED either in TMA-0 >= 2 at (A) and TMA-24 >= 3 (B), while the mortality percentage during later time course (Day 5 to 30) seems less significant than the control. Thus, this reviewer doubted whether TAM is merely a predictor of acute mortality of septic shock. Please discuss.
Response: Thank you for your important comment. We have discussed this point with our statisticians. Based on the optimal cutoff values of 2 for TMA on ED admission and of 3 for TMA at time-24, Kaplan-Meier survival curves were created using 7-day and 30-day mortality data, and the groups were compared using the log-rank test. Consequently, higher TMA scores at admission and 24 hours after admission were significantly associated with an increased risk of 30-day mortality among patients with septic shock.
Supplement 4. The TMA score as a predictor of 30-day mortality. Higher TMA score at admission (A) and 24 h (B) after admission were significantly associated with an increased risk of 7-day mortality among patients with septic shock.
In addition, as you commented, we also analyzed effects of the TMA score over the short-term. TMA score time-0 and time-24 were also significantly associated with an increased risk of 7-day mortality among patients with septic shock. We have added the statistical significances of TMA score time-0 and time-24 in the development of mortality within 7 days to supplement S4 as figures; this is because most other studies have used primarily mortality events within 7 days rather than 5 days. In addition, we analyzed mortality events within 30 days with the assistance of our statisticians.
In contrast to the TMA time-0 < 2 group, TMA time-0 ≥ 2 group showed a higher and consistent occurrence of mortality events after 5 days. The incidence of the mortality events after 5 days was higher in the TMA time-0 ≥ 2 group than in the TMA time-0 < 2 group.
In the TMA score time-24 ≥ 3 and < 3 groups, the frequency of events after 5 days was similar. The similarity of the incidence of death after 5 days between the two TMA time-24 groups seems to be because overall event was as relatively small as just 31. The data were limited owing to the small size of the sample in a single, tertiary, academic hospital. Further prospective, multicenter trials are required to validate the usefulness of the TMA scores as a prognostic marker. However, the TMA time-0 ≥ 2 group showed a consistent and higher occurrence of mortality events after 5 days. First, since many studies concerning sepsis often refer to the occurrence of mortality events within 30 days as a point of interest, we also determined the events within 30 days as a main variable for relative comparison with other studies. There is a significant difference in the cause of death according to the timing of death in patients with severe sepsis/septic shock, i.e., early death (within 3 days) and late death. Sudden cardiovascular collapse or multi-organ failure is the most common cause of early death, and secondary complications, including nosocomial infections, are known to be responsible for late death. We wanted to use integrative outcomes that include both early death and late death in patients with severe sepsis/septic shock. Based on comments of reviewers and our statisticians, in the future, we will consider further prospective, multicenter trials considering the number of event occurrences.
Daviaud, F., et al. (2015). "Timing and causes of death in septic shock." Ann Intensive Care 5(1): 16.
Hernandez, G., et al. (2019). "Effect of a Resuscitation Strategy Targeting Peripheral Perfusion Status vs Serum Lactate Levels on 28-Day Mortality Among Patients With Septic Shock: The ANDROMEDA-SHOCK Randomized Clinical Trial." JAMA 321(7): 654-664.
Sheth, M., et al. (2019). "The association between autoimmune disease and 30-day mortality among sepsis ICU patients: a cohort study." Crit Care 23(1): 93.
Rivers, E., et al. (2001). "Early goal-directed therapy in the treatment of severe sepsis and septic shock." N Engl J Med 345(19): 1368-1377.
2) The author highlighted the “independence” of TAM in predicting 30-day mortality both in abstract and main text. However, the author also used TAM combined with other variables e.g. SOFA or lactate to compare the sensitivity and specificity, especially when add to lactate levels could significantly increase the predictability. Thus, from the future practically perspective, such highlight is not necessary.
Response: Thank you for your comment. We also agree with your comment. Accordingly, we have removed the term “independence” from the abstract and main text. In addition, considering the future practical perspective, we have added this point to the discussion of the manuscript.
The TMA score in the ED may be useful to predict 30-day mortality in patients with severe sepsis and septic shock. Combining TMA score with lactate concentrations improves the predictive performance for the prognosis and may contribute to rapid risk stratification of patients with severe sepsis and/or septic shock.
3) Page 4, line 185. The author cited table 1 only once in the main text with only a rough description (line 184-185). Actually, there are more than TAM showed statistic significance in Table one, e.g. cardiovascular disease, albumin, lactate, total CO2. Please describe the date in the main text more detail and discuss if necessary, while please do not leave the readers to guess where the description for these data are.
Response: Thank you for your comment. We also agree with your comment. Accordingly, we have added this point to the results of manuscript.
Ninety-one of them were male (44.61%), and the mean age was 67 ± 15 years. The mortality group (29.03%) showed higher rates of cardiovascular disease than did the survival group (13.87%, p=0.035). The SOFA score was significantly increased in patients who died within 30 days (10.39 ± 2.93), as compared with patients who survived (7.11 ± 2.53,<0.001)< span="">. There were significant differences for lactate, albumin, and total CO2 between the two groups (p of all<0.001). TMA score at time-0 and -24 were significantly increased in patients who died as compared to patients who survived (1.42 ± 1.02 and 1.69 ± 1.16 in the survival group versus 2.23 ± 1.15 and 2.82 ± 1.30 in the mortality group, p of all <0.001). Mean admission to antibiotics time was 3.64 h (SD 4.54), and this was significantly longer in patients who died within 30 days compared to those who did not (3.81 ± 4.88 versus 2.70 ± 1.44, p=0.016).
4) Page 7, table 2. First, it is not clear what’s the difference between the left half panel and right half panel, because the top parameters are the same. Second, the format needs to be modified, the p value should be in the same line, it’s quite ambiguous now.
Response: Thank you for your important comment. We have discussed this point with our statisticians. Since there was multi-collinearity between TMA time-0 and -24, we could not consider both TMA time-0 and -24 together in one model. Each TMA value can be used as a meaningful predictor on ED admission and 24 hours after admission. Therefore, we analyzed it by separating it into two models with TMA-0 and -24. In addition, the enter method, considering clinical and statistical characteristics, was intended to determine whether TMA-0 and -24 are significant even after adjusting for important variables. Unlike the enter method, the stepwise method considering only statistical characteristics determined whether TMA-0 and -24 were significant even after adjusting for statistically important variables.
To avoid confusion among readers, we modified the table by separating models including TMA score time-0 and -24 by the enter method and stepwise method.
5) Please explain why the ROC curve of TAM-0 in B and TAM-24 in C where totally opposite with the one in Figure 4.
Response: Thank you for your important comments. Based on your comment, we have corrected the ROC curves of TMA-0 and -24 in Figure 4.
6) Page 9, line 240-242. “The AUC of the addition of the TMA score at time-0 and time-24 to the lactate level (0.8; IQR, 0.703-0.878)” this value should be replaced by the later one in line 242.
Response: Thank you for your important comments. Based on your comment, we have revised this point below.
To improve the predictability of clinical outcomes early after ED admission, we demonstrated how predictability was improved when the TMA score was added to the lactate level on ED admission. The AUC of the addition of the TMA score at time-0 or time-24 to the lactate level (0.8; IQR, 0.703-0.878) on ED admission was significantly improved compared to the AUC of the lactate level alone (0.857 [p=0.034] and 0.892 [p=0.018])
7) In the conclusion, the authors indicated that it can be helpful when determine the initial treatment strategies with the use of TAM scores. However, no such discussions were presented. What will the initial treatment strategies be different with TAM scores in the future?
Response: Thank you for your comment. We discussed this point with all authors. Considering your comment, we may have overstated the future usefulness of TMA. We have revised this point in the conclusion.
We found that increased TMA scores predicted the 30-day mortality of patients with severe sepsis and septic shock. Combining TMA score with initial lactate concentrations can improve the predictive performance of short-term mortality. The TMA score is automatically measurable by an automated hematology analyzer. It can be quickly measured without additional costs or effort; therefore, the TMA score may be a promising tool for rapid risk stratification of patients with severe sepsis and/or septic shock in need of intensive care and innovative therapies.
Minor issues:
1) Page 2, line 82-85, lack of reference after “lactate levels have been used clinically as a reliable indicator … using point-of care testing”.
Response: Based on your comment, we have added the following references.
1. Gomez H, Kellum JA. Lactate in sepsis. Jama. 2015;313(2):194-5.
2. Casserly B, Phillips GS, Schorr C, Dellinger RP, Townsend SR, Osborn TM, et al. Lactate measurements in sepsis-induced tissue hypoperfusion: results from the Surviving Sepsis Campaign database. Critical care medicine. 2015;43(3):567-73.
2) Since the author compared TMA score with CRP in line 224-225, page 8. Please add CRP ROC cure in Figure 4.
Response: According to your comment, we have revised the presentation of ROC for CRP in Figure 4.
The revised manuscript has been proofread by a native English editor from the Department of Research Affairs, Yonsei University College of Medicine, and we have availed English language scientific editing services provided by Editage of Cactus Communications, Inc.
Reviewer 2 Report
I'm really impressed about the value of this paper. Congratulations for Authors. The abstract is very readable and well presents the content of the paper. All other parts are also prepared with high scientific and clinical standards. I have only minor suggestions:
In abstract, please delete the numbers
Line 58 editorial error
In table 1, please change the format because in a few cases the p-value is not readable.
Author Response
Type of the Paper (Article)
Usefulness of the Thrombotic Microangiopathy Score as a Promising Prognostic Marker of Septic Shock for Patients in the Emergency Department
Response to reviewer #2’s comments
Comments and Suggestions for Authors
I'm really impressed about the value of this paper. Congratulations for Authors. The abstract is very readable and well presents the content of the paper. All other parts are also prepared with high scientific and clinical standards. I have only minor suggestions:
Response: Thank you for these comments and useful suggestions regarding our manuscript. We have responded to the comments in detail below.
In abstract, please delete the numbers
Response: According to your comment, we have revised the abstract.
Line 58 editorial error
Response: According to your comment, we have checked this point.
In table 1, please change the format because in a few cases the p-value is not readable.
Response: According to your comment, we have revised to address this issue.
The revised manuscript has been proofread by a native English editor from the Department of Research Affairs, Yonsei University College of Medicine, and we have availed English language scientific editing services provided by Editage of Cactus Communications, Inc.
Round 2
Reviewer 1 Report
This meta-analysis manuscript has been thoroughly revised and the quality has been improved. There are some minor issues still need to be corrected:
1) Please add Table S4 in line 594 of revised version. Make sure the title of this table is correct because in the response letter, it was wrongly labeled as “The TMA score as a predictor of 30-day mortality”.
2) The author added detailed description of Table 1 results in the main text, which is much better now. However, please check the line 380-381, the data of “Mean admission to antibiotics time” was cited completed opposite with the one shown in the Table, where the patients die in 30-day was shorter than patients who did not.
3) The author had not corrected the mistake in my question #6. The AUC value was not exchanged In line 459-461, which was cited at the wrong place.
Author Response
Type of the Paper (Article)
Usefulness of the Thrombotic Microangiopathy Score as a Promising Prognostic Marker of Septic Shock for Patients in the Emergency Department
Response to reviewer #1’s comments
Comments and Suggestions for Authors
This meta-analysis manuscript has been thoroughly revised and the quality has been improved. There are some minor issues still need to be corrected:
Response: Thank you for these comments and useful suggestions regarding our manuscript. We are really sorry for our mistakes. We have responded to the comments in detail below.
1) Please add Table S4 in line 594 of revised version. Make sure the title of this table is correct because in the response letter, it was wrongly labeled as “The TMA score as a predictor of 30-day mortality”.
Response: Thank you for your important comment. Accordingly, we have added this point to the revised manuscript.
Added this point to the manuscript
Table S4: The TMA score as a predictor of 7-day mortality
Confirmed the title in S4
Supplement 4. The TMA score as a predictor of 7-day mortality
2) The author added detailed description of Table 1 results in the main text, which is much better now. However, please check the line 380-381, the data of “Mean admission to antibiotics time” was cited completed opposite with the one shown in the Table, where the patients die in 30-day was shorter than patients who did not.
Response: Thank you for your important comment. Accordingly, we have revised this point to the manuscript.
Mean admission to antibiotics time was 3.64 h (SD 4.54), and this was significantly longer in patients who died within 30 days compared to those who did not (3.81 ± 4.88 versus 2.70 ± 1.44, p=0.016).
Mean admission to antibiotics time was 3.64 h (SD 4.54), and this was significantly shorter in patients who died within 30 days compared to those who did not (2.70 ± 1.44 versus 3.81 ± 4.88, p=0.016).
3) The author had not corrected the mistake in my question #6. The AUC value was not exchanged In line 459-461, which was cited at the wrong place.
Response: Thank you for your important comment. Accordingly, we have revised this point to the manuscript.
To improve the predictability of clinical outcomes early after ED admission, we demonstrated how predictability was improved when the TMA score was added to the lactate level on ED admission. The AUC of the addition of the TMA score at time-0 or time-24 to the lactate level (0.8; IQR, 0.703-0.878) on ED admission was significantly improved compared to the AUC of the lactate level alone (0.857 [p=0.034] and 0.892 [p=0.018])
To improve the predictability of clinical outcomes early after ED admission, we demonstrated how predictability was improved when the TMA score was added to the lactate level on ED admission. The AUC of the addition of the TMA score at time-0 or time-24 to the lactate level (0.857 [p=0.034] and 0.892 [p=0.018]) on ED admission was significantly improved compared to the AUC of the lactate level alone (0.8; IQR, 0.703-0.878).
We appreciate your comments and useful suggestions.